# Effect of Plano-Valgus Foot on Lower-Extremity Kinematics and Spatiotemporal Gait Parameters in Children of Age 5–9

**DOI:** 10.3390/diagnostics12010002

**Published:** 2021-12-21

**Authors:** Anna Boryczka-Trefler, Małgorzata Kalinowska, Ewa Szczerbik, Jolanta Stępowska, Anna Łukaszewska, Małgorzata Syczewska

**Affiliations:** Department Rehabilitation, The Children’s Memorial Health Institute, Al. Dzieci Polskich 20, 04-730 Warszawa, Poland; a.boryczka-trefler@ipczd.pl (A.B.-T.); m.kalinowska@ipczd.pl (M.K.); e.szczerbik@ipczd.pl (E.S.); jolanta@stepowscy.pl (J.S.); a.lukaszewska@ipczd.pl (A.Ł.)

**Keywords:** children, flat foot, gait, classification, statics vs. dynamics

## Abstract

Aim of the study was to see how a definition of the flexible flat foot (FFF) influences the results of gait evaluation in a group of 49 children with clinically established FFF. Objective gait analysis was performed using VICON system with Kistler force platforms. The gait parameters were compared between healthy feet and FFF using two classifications: in static and dynamic conditions. In static condition, the ink footprints with Clarke’s graphics were used for classification, and in dynamic condition, the Arch Index from Emed pedobarograph while walking was used for classification. When the type of the foot was based on Clarke’s graphics, no statistically significant differences were found. When the division was done according to the Arch Index, statistically significant differences between flat feet and normal feet groups were found for normalized gait speed, normalized cadence, pelvic rotation, ankle range of motion in sagittal plane, range of motion of foot progression, and two parameters of a vertical component of the ground reaction force: FZ2 (middle of stance phase) and FZ3 (push-off). Some statically flat feet function well during walking due to dynamic correction mechanisms.

## 1. Introduction

A plano-valgus foot is the most common posture deformity among children [1,2,3]. Despite this fact, there are still neither unambiguous diagnostic criteria of pediatric plano-valgus foot nor commonly agreed foot assessment methods. That is why the prevalence of pediatric flexible flat foot in the literature is rated from a few to ten per cent, and it depends very much on diagnostic methods used, their accuracy, evaluation criteria, children’s age, their gender, and weight [4,5,6,7,8]. Assessment methods used by clinicians vary from clinical observation to measurements and imaging techniques both in weight-bearing and no weight-bearing positions or in static and dynamic conditions. Still, the reliability, validity, and accuracy of all these methods are unproven [9,10,11]. In our previous study [12], when the same feet were assessed using two different methods, one in static condition and one while walking, we found out the significant difference between the classification outcome: 35 feet (out of 100) classified as flat by static method were not flat according to dynamic classification method, and four feet classified as normal according to static method were flat according to the dynamic method.

Nowadays, the plano-valgus foot kinematics assessment methods are gaining importance because they can be used not only for the evaluation of the flat foot posture but also for the assessment of the flat foot performance during walking. Additionally, the influence of the flat foot on the overall gait pattern can be assessed. They seem to be more objective and their results more compatible among researchers. Twomey et al. [13] found increased forefoot supination and medial longitudinal arch (MLA) collapse during walking in children with a flat foot; Caravaggi et al. [14] reported greater hindfoot eversion and its plantarflexion relatively to the tibia, larger MLA collapse, and hallux dorsiflexion throughout most of the stance phase, dorsiflexion, eversion, and abduction of the midtarsal joint and plantarflexion and adduction of the tarso-metatarsal joint. He did not observe any significant forefoot abduction relatively to the hindfoot. Similar to Caravaggi, Saraswat et al. [15] also observed larger hindfoot eversion and plantarflexion together with increased midfoot pronation and dorsiflexion in plano-valgus foot. Similar observation concerning plano-valgus foot during gait was also made by Kerr et al. [16], Kothari et al. [17], and others [18,19].

The human musculoskeletal system is a biomechanical chain; therefore, a pertinent question is whether the plano-valgus foot deformity affects not only foot joints but also upper joints of the lower extremities, pelvis. and lower back [19]. Duval et al. [20] observed that placing a foot in eversion caused subtalar pronation and this resulted in the increased internal knee and hip rotation, while placing a foot in inversion resulted in subtalar supination and increased external knee and hip rotation. However, he did not find any evidence of dependence between increased foot pronation or supination and pelvic anterior or posterior tilt. Opposite results were obtained by Pinto et al. [21]. He stated that both unilateral and bilateral calcaneal eversion obtained using medially tilted wedges resulted in pelvic anteversion. Additionally, unilateral calcaneal eversion caused a lateral pelvic tilt. Svoboda et al. [19] also reported an increase in pelvic anteversion as a result of unilateral and bilateral hindfoot eversion and additionally a significantly higher hip external rotation during the first half of the stance phase with bilateral everted hindfoot. Additionally, a study of Lopez and co-workers [22] found that the foot arch height has a global, negative impact on the quality of life of the schoolchildren, proving the importance of the foot deformities on the overall wellbeing. The similar study done under the same leadership [23] in the adults did not show any dependence between the height of the foot arch and quality of life although another study performed in the adults with foot pathologies showed that they have a worse quality of life than the general population [24].

Taking into consideration the wide range of clinical diagnostic tools and findings concerning the influence of the flat foot on gait pattern, the aim of this study was to see how a definition of the flexible flat foot (FFF) influences gait parameters in children five to nine years of age and if the choice of a diagnostic method of FFF used in the study (in static vs. dynamic conditions) affects its results. The definitions of FFF used in practice differ from each other, which means that the applied method of foot classification influences the assessment of the patient’s gait stereotype and the resulting therapeutic management. The importance of the research undertaken is due to the potentially negative impact of foot deformation on the quality of life in adulthood.

## 2. Materials and Methods

### 2.1. Patients

Forty-nine children (37 boys and 12 girls) were recruited to the study. Recruitment was carried out in the period of two years during the clinical examination at The Children’s Memorial Health Institute in Warsaw, Dept. Rehabilitation, at the Outpatient Clinic. All children fulfilling the criteria were invited to participate. The inclusion criteria were as follows: age from 5 to 9 years and flexible flat foot, clinically established. The exclusion criteria were: rigid flat foot, secondary flat foot caused by the damaged central nervous system (CNS), neuromuscular diseases, lower-limb injury, or surgical intervention in the lower legs in the past. The demographic characteristic of the group is presented in Table 1. The study was approved by the Local Ethical Committee. It was a prospective cohort type study.

Informed consent was obtained from the parents of all children taking part in the study before their enrolment.

### 2.2. Methods

Figure 1 presents the flow chart of the study.

### 2.3. Clinical Feet Assessment

Preliminary diagnose of the flexible flat foot was based on a clinical examination conducted independently by an experienced physician and physiotherapist. A foot was defined as flat when, during the examination while standing, the MLA was collapsed, and/or the medial side of the foot was bulging because of the talus head protruding just under the medial malleolus. The heel valgus angle was measured with a goniometer during standing on both feet. It was measured three times, and then, an averaged result was calculated. Flexible flat foot was identified when the MLA rebuilt in non-weight-bearing position and while tiptoe standing.

### 2.4. Ink Footprints

After the preliminary examination, ink footprints from the Harris and Beath pedograph were obtained, and they were further compared with Clarke’s footprinting graphics. A foot was diagnosed as flat if the ink footprint from the Harris and Beath pedograph matched Clarke’s footprinting graphics types between 7 and 10. The matching of footprints was performed independently by two experienced examiners, and no discrepancy between their results occurred. The complete description of the examination methodology on the Harris and Beath pedograph is included in the previous study [12].

### 2.5. Pedobarography

Next, plantar loads during gait were evaluated (Figure 2A). Plantar loads were captured using the emed system (Novel Company) [25]. The complete description of the examination methodology on the emed platform is included in a previous study [12].

Data from three plantar loads of left and three plantar loads of right foot of each child were averaged and taken for further analysis. Geometric measures of the feet (midfoot width, instep width, instep, foot width) were calculated by Novel software. The Arch Index was calculated based on the definition introduced by Cavanagh and Rodgers. The value of Arch Index equal to 0.27 was taken as cut-off value between normal and flat foot.

### 2.6. Instrumented Gait Analysis

Objective gait analysis was performed using a 12 camera VICON MX System (Figure 2B). The Plug-In-Gait marker set and lower-body model were used. Patients walked with their preferred, self-selected speed several times along the walkway to obtain six technically correct trials, which were later imported to the Polygon software and averaged. The data extracted from the averaged reports were later analysed. Spatio-temporal data were expressed as per cent of the age- and sex-matched reference data [26]. The following parameters were taken into the analysis: gait speed, cadence, step length, step width, stance phase, single-stance phase, pelvic tilt, pelvic range of motion (ROM) in sagittal plane, pelvic obliquity, pelvic range of motion in frontal plane, pelvic rotation, pelvic range of motion in transversal plane, hip flexion at initial contact, hip flexion in terminal stance, hip flexion in swing, pass retract, hip range of motion, hip abduction, hip range of motion in frontal plane, hip rotation in swing, hip range of motion in transversal plane, knee flexion at initial contact, knee flexion in weight acceptance, knee flexion in standing, maximal knee flexion in swing, knee flexion in terminal swing, knee range of motion, dorsiflexion at initial contact, maximal dorsiflexion in standing, maximal plantarflexion, plantarflexion in swing, ankle range of motion in sagittal plane, foot progression, range of motion in foot progression, and maximal values of ground reaction force components (vertical, medio-lateral, and fore-aft).

### 2.7. Statistical Analysis

Statistical analysis of the data comprised a chi-square test to check the variables’ distribution, the Wilcoxon signed-rank test for matched pairs to find differences between parameters of left and right leg, and the Mann–Whitney U test to compare lower extremities ROM and kinematic and spatio-temporal parameters of the healthy and flat feet. The statistically significant level was set as 0.05.

Two different comparisons were performed in the analysis. In the first one, the data were divided into two groups based on Clarke’s classification: the first group consisted of flat feet and the second of normal feet. In the second one, the division to flat and normal feet was based on the Arch Index from dynamic walking on emed platform.

## 3. Results

The comparisons of the parameters between left and right leg, done with Wilcoxon signed-rank test, showed no differences; thus, the data from left and right legs were pooled together.

When the type of the foot was based on Clarke’s footprinting graphics, no statistically significant differences were found between flat feet and normal feet groups in spatio-temporal, kinematic, or ground reaction force parameters.

In the second case, when the division was done according to the Arch Index from the pedobarography, the following parameters were statistically significantly different between flat feet and normal feet groups: normalized gait speed (Figure 3), normalized cadence (Figure 3B), pelvic rotation (Figure 4A), ankle range of motion in sagittal plane (Figure 4B), range of motion of foot progression (Figure 4C), and two parameters of a vertical component of the ground reaction force: FZ2 (middle of stance phase) (Figure 5A) and FZ3 (push-off) (Figure 5B).

The summary statistics of all analysed parameters is given in Table 2 and Table 3. Table 2 presents the parameters when the feet were divided into flat and normal feet groups according to the ink footprinting and Clarke’s definition and Table 3 when the feet were divided according to the Arch Index from pedobarography. As all the parameters were non-normally distributed (as showed by the results of the chi-square test), the data were summarized by the medians and 10th and 90th percentiles.

## 4. Discussion

It is commonly believed that flat foot affects walking pattern [14,27]. Although some tests involving children’s sport performance showed no difference between children with and without flat foot, the clinical observations show that a great part of flat feet are symptomatic, and more and more researchers find proof that not just symptomatic, but also asymptomatic flat feet do affect function [14,19,28,29]. Such discrepancies between the researchers may be a consequence of different diagnostic methods they use to classify a flat foot for their research. That is why it is also so difficult to compare different study results.

The aim of this study was to investigate how, if at all, a FFF influences gait parameters in children and if a choice of a diagnostic method used to identify FFF affects the results of the gait pattern assessment. We examined spatio-temporal, kinematic, and kinetic parameters of the flat feet and healthy feet in a group of children, using two different classification methods. The main result is the finding that a diagnostic method according to which the flat foot is established has an important impact on the results. The statistically significant differences of gait parameters between healthy and flat feet were found only when the classification was based on the Arch Index in dynamic condition. We decided to use two classification methods because defects in foot posture in static conditions are not always seen in dynamic conditions: in fact, flat foot posture is not always accompanied by the impaired function [12,18,30,31]. In our previous study, it was proven that there is a significant difference between the outcome when classifying the feet in static and dynamic conditions [12]. A great number of feet classified in static conditions as flat feet according to the classification executed in dynamic conditions turned out to be not flat.

Examining the spatio-temporal parameters, we found, as observed also by Carravaggi et al., Lin et al., and Hösl et al. [7,14,18] a statistically significant decrease in walking speed and cadence in children with flat feet in comparison to healthy feet. Lin and co-authors additionally observed a reduction in stride length, which was not the case in our study [7]. Similar results but in adults were found by Levinger et al. [31]. He found a reduction in cadence but, contrary to Lin’s study, an increase in stride length.

From other researchers’ studies, it is already known that speed is a factor that significantly affects both kinematic and kinetic parameters, such as joint ROM, joints moments, the ground reaction forces [32]. Stansfield et al. in his longitudinal study of gait of healthy children (5–12 years old) stated that walking speed has a greater impact on gait parameters than age [33,34]. He found that a decreased walking speed can cause the decrease in the peak plantar flexion angle.

Regarding the kinematic parameters in this study, a statistically significant decrease in ankle ROM in sagittal plane was observed in children with FFF in comparison to healthy feet. The decrease in the ankle range of motion in a sagittal plane means a weaker push-off during gait and relates to a lower FZ3—a parameter of the vertical component (second maximum) of the ground reaction force during this phase of gait. Similar results were obtained by other researchers [18,31]. Hösl et al. [18] observed a limited hindfoot motion in the sagittal plane, which was probably compensated by increased midfoot dorsiflexion and an excessively mobile hallux during the push-off phase. He also noticed a trend towards lower FZ3 in the symptomatic flat foot together with a reduced gait speed. Remarkably similar results were obtained by Saraswat et al. [15] He observed a reduced ROM in the sagittal plane of an ankle joint in children with flat feet, accompanied by its eversion and plantarflexion. Regarding kinetic parameters, the smaller plantarflexion and outward rotation moment peaks together with smaller power generated by an ankle joint of the FFF were found.

Recently more proofs were found to support the hypothesis that morphology of the flat foot is not always accompanied by its abnormal function [31]. Therefore, maybe we should differentiate between morphological features of flat foot and its influence on the function, i.e., walking. That is why, in our study, we used two methods of flat foot classifications: in static and in dynamic conditions. Using the classification in static conditions, we did not find any statistically significant differences between flat and healthy feet in any functional parameters, i.e., spatio-temporal, kinematic, and kinetics parameters. This finding can lead to the conclusion that examining foot posture in static conditions does not help a clinician to find patients who have real functional walking problems. Sometimes statically flat feet function well during walking because they have the potential of dynamic correction of themselves. Thus, maybe a clinical examination in static conditions should not be the only one while deciding on the treatment. It seems that the dynamic tests, which identify individuals with functional problems, should be the basis for planning the treatment. Children with FFF identified in static conditions who do not have gait impairments should probably be put under observation and not immediately under treatment. A classification done in dynamic conditions identifies children with FF who have walking impairments and really need treatment.

The main limitation of the present study is the relatively low number of patients and the imbalance between female and male participants. This resulted from the fact that the patients were recruited from the outpatient clinic, and all patients who fulfilled the criteria were invited to participate.

In conclusion, the diagnosis of the flat foot based on the evaluation in the static condition and during the clinical assessment seems not be sufficient for decision making about the treatment of pediatric patients with flexible flat foot. One of the main findings from our study is that the gait pattern pathology seen in the gait parameters can depend on the classification method within the same group of patients with clinical problem.

## Figures and Tables

**Figure 1 diagnostics-12-00002-f001:**
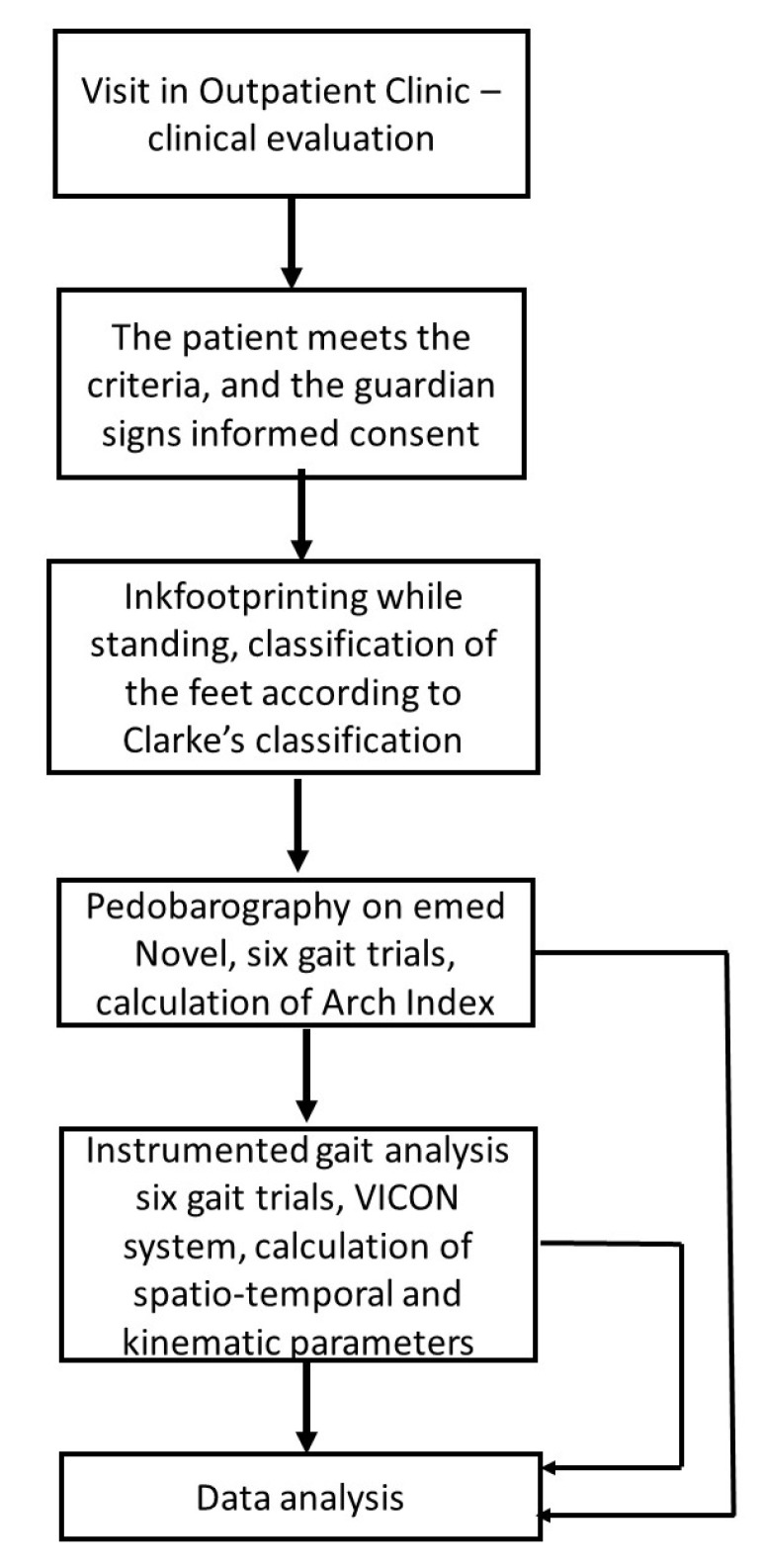
Flow chart of the study.

**Figure 2 diagnostics-12-00002-f002:**
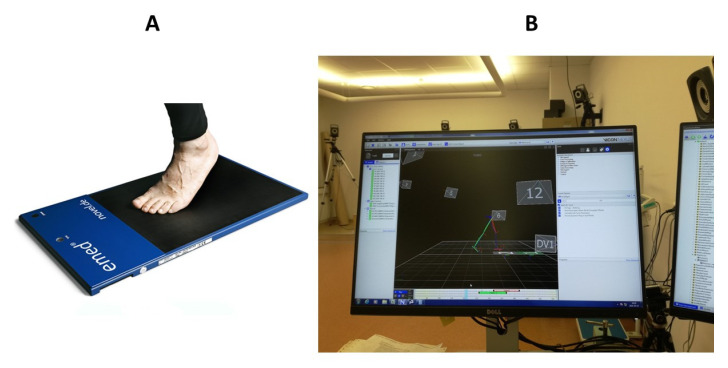
(**A**) Gait trial during pedobarography (photo from Novel’s web page www.novel.de, accessed on 28 November 2021). (**B**) Instrumented gait analysis.

**Figure 3 diagnostics-12-00002-f003:**
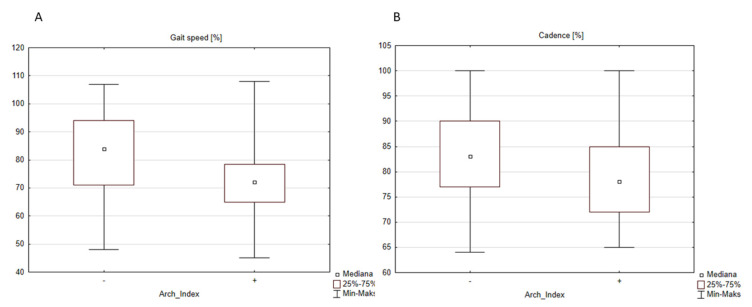
The influence of the type of the foot defined by the Arch Index on the spatio-temporal parameters: (**A**) speed, “-”—normal foot (median = 84.0%), “+” flat foot (median = 72.0%), and (**B**) cadence, “-” normal foot (median = 83.0%), “+” flat foot (median = 78.0%).

**Figure 4 diagnostics-12-00002-f004:**
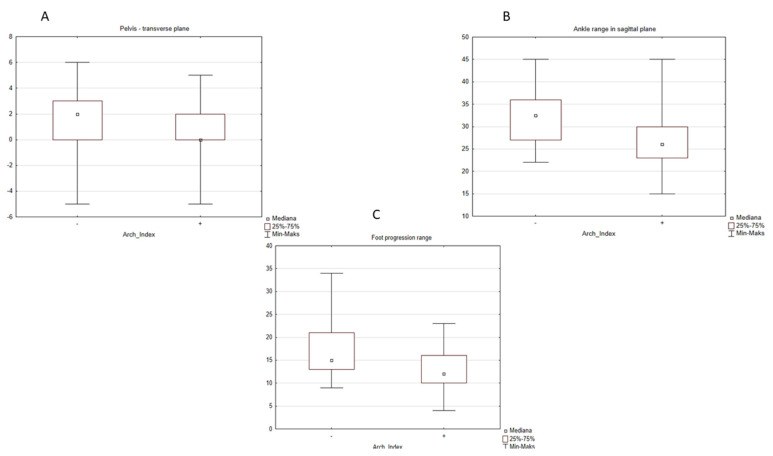
The influence of the type of the foot defined by the Arch Index on the kinematics: (**A**) pelvic rotation in transverse plane, “-” normal foot (median = 2.0), “+” flat foot (median = 0.0). (**B**) Ankle range in sagittal plane, “-” normal foot (range = 32.5), “+” flat foot (median = 26.0), and (**C**) the foot progression range, “-” normal foot (median = 15.0), “+” flat foot (median = 12.0).

**Figure 5 diagnostics-12-00002-f005:**
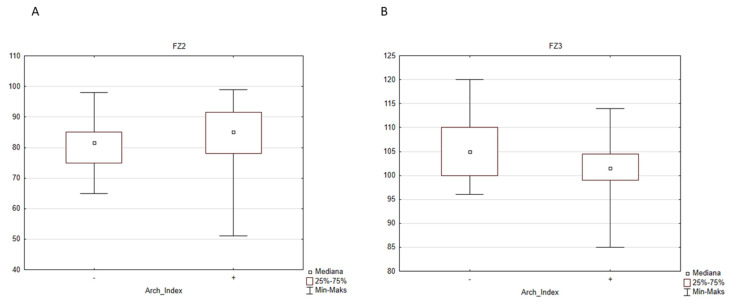
The influence of the type of the foot defined by the Arch Index on the parameters of the vertical ground reaction force: (**A**) FZ2 (middle of stance phase), “-” normal foot (median = 81.5% BW), “+” flat foot (median = 85.0% BW), and (**B**) FZ3 (push-off), “-” normal foot (median = 105.0% BW), “+” flat foot (median = 101.5% BW). BW, body weight.

**Table 1 diagnostics-12-00002-t001:** Demographic characteristics of the group.

	No of Subjects	Median	Minimum	Maximum	10th Percentile	90th Percentile
Height	49	124.5	109.5	140.0	113.0	135.0
Body mass	49	24.0	18.7	39.0	20.5	34.6
BMI	49	16.23	12.62	21.73	14.22	19.19
Age	49	6.41	5.04	10.37	5.24	8.20

**Table 2 diagnostics-12-00002-t002:** The gait parameters when the feet were divided according to the ink footprinting. The parameters were summarized by medians and 10th and 90th percentiles. Z—the results of the Mann-Whitney U test, *p*-level—probability value.

Parameter	Flat Foot Group	Normal Foot Group	Z	*p*-Level
Normalized gait speed (%) *	75.0 <56.0–100.0>	74.5 <61.5–104.0>	−0.333	0.739
Cadence (%) *	79.0 <68.0–92.0>	81.0 <71.0–94.5>	−0.369	0.711
Step width (m)	0.16 <0.12–0.2>	0.155 <0.13–0.235>	−0.454	0.649
Normalized step length (%) *	93.0 <79.0–115.0>	95.5 <84.0–115.0>	−0.916	0.360
Stance phase (%)	60.6 <58.8–63.2>	61.7 <58.3–64.1>	−0.734	0.463
Single stance phase (%)	39.5 <35.0–43.0>	39.2 <36.6–44.4>	0.006	0.995
Pelvic tilt (deg)	8.0 <2.0–15.0>	10.0 <−0.5–15.5>	−0.340	0.734
Pelvic range in sagittal plane (deg)	3.0 <2.0–5.0>	3.0 <2.0–5.0>	−0.261	0.794
Pelvic obliquity (deg)	0.0 <−3.0–2.0>	0.0 <−3.5–3.0>	0.024	0.981
Pelvic range in frontal plane (deg)	6.0 <5.0–10.0>	6.0 <5.5–8.5>	−0.455	0.649
Pelvic rotation (deg)	0.0 <−2.0–5.0>	0.0 <0.0–4.0>	−0.366	0.714
Pelvic range in transverse plane (deg)	9.0 <5.0–14.0>	8.5 <4.5–12.5>	0.540	0.589
Hip flexion at initial contact (deg)	24.0 <15.0–34.0>	26.5 <16.5–36.5>	−0.309	0.757
Hip flexion at terminal stance (deg)	−13.0 <−22.0–−3.0>	−11.0 <−25.5–−5.5>	−0.170	0.865
Hip flexion in swing (deg)	26.0 <17.0–36.0>	28.0 <18.5–40.5>	−0.449	0.654
Pass retract (deg)	0.0 <0.0–5.0>	3.0 <0.0–5.0>	−0.914	0.361
Hip range in sagittal plane (deg)	38.0 <31.0–46.0>	38.5 <33.5–49.0>	−0.285	0.776
Hip range in sagittal plane (%)	90. 0 <76.0–105.0>	92.0 <77.5–116.5>	−0.358	0.721
Hip abduction (deg)	0.0 <−5.0–5.0>	−1.0 <−5.0–4.5>	0.667	0.505
Hip range in frontal plane (deg)	10.0 <6.0–13.0>	10.0 <7.0–13.5>	−0.164	0.870
Hip rotation (deg)	−8.0 <−22.0–14.0>	−7.5 <−15.0–13.5>	−1.644	0.100
Hip range in transverse plane (deg)	20.0 <14.0–35.0>	19.0 <13.5–34.0.	0.434	0.664
Knee flexion at initial contact (deg)	0.0 <−4.0–5.0>	1.0 <−2.5–8.0>	−0.819	0.413
Knee flexion at weight acceptance (deg)	11.0 <5.0–17.0>	11.5 <5.5–21.0>	−0.400	0.689
Knee flexion at midstance (deg)	1.0 <−4.0–5.0>	0.5 <−4.5–7.0>	−0.182	0.856
Max knee flexion at swing (deg)	53.0 <46.0–58.0>	54.0 <48.5–60.0>	−0.673	0.501
Knee flexion in terminal swing (deg)	−5.0 <−11.0–−2.0>	−5.0 <−6.0–−3.0>	−0.772	0.440
Knee range in sagittal plane (deg)	55.0 <47.0–62.0>	55.0 <47.0–64.0>	−0.461	0.645
Ankle flexion at initial contact (deg)	−5.0 <−10.0–0.0>	−4.5 <−9.0–2.0>	0.018	0.985
Max dorsiflexion in swing (deg)	14.0 <8.0–17.0>	14.5 <9.5–18.5>	−0.606	0.544
Max plantarflexion (deg)	−13.0 <−28.0–−2.0>	−17.0 <−20.0–−2.0>	0.434	0.664
Ankle range in sagittal plane (deg)	27.0 <22.0–39.0>	29.0 <21.0–33.5>	−0.109	0.914
Foot progression (deg)	−3.0 <−15.0–8.0>	0.0 <−8.0–13.5>	−0.806	0.420
Range of foot progression (deg)	13.0 <9.0–22.0>	12.5 <9.5–17.0>	0.327	0.743
FZ1 **	104.0 <94.0–120.0>	101.0 <93.5–114.5>	0.891	0.373
FZ2 **	85.0 <71.0–94.0>	82.5 <66.5–94.0>	0.509	0.611
FZ3 **	102.0 <95.0–110.0>	103.0 <98.5–115.5>	−0.685	0.493
FX1 **	9.0 <7.0–13.0>	9.0 <8.0–11.0>	−0.200	0.841
FX2 **	0.0 <0.0–2.5>	0.3 <0.0–3.8>	−0.382	0.702
FY1 **	18.0 <12.0–23.0>	16.5 <11.0–26.0>	0.315	0.753
FY2 **	18.0 <12.0–24.0>	18.0 <14.5–25.0>	−0.806	0.420

* normalized to age matched reference data of healthy children; ** normalized to body weight.

**Table 3 diagnostics-12-00002-t003:** The gait parameters when the feet were divided according to the Arch Index from pedobarograhy. The parameters were summarized by medians and 10th and 90th percentiles. The statistically significant differences were marked by bolded font. **Z—the results of the Mann-Whitney U test, *p*-level—probability value**.

Parameter	Flat Foot Group	Normal Foot Group	Z	*p*-Level
**Normalized gait speed (%) ***	**72.0 <56.0–102.0>**	**84.0 <61.0–100.0>**	**2.112**	**0.035**
**Cadence (%) ***	**78.0 <68.0–92.0>**	**83.0 <72.0–94.0>**	**2.243**	**0.025**
Step width	0.16 <0.12–0.22>	0.15 <0.13–0.19>	−1.373	0.170
Normalized step length (%) *	90.0 <77.0–113.0>	96.0 <80.0–116.0>	−1.699	0.089
Stance phase (%)	61.0 <59.0–63.8>	60.4 <58.7–62.9>	−0.701	0.483
Single stance phase (%)	39.2 <35.0–43.4>	39.8 <35.5–43.0>	0.430	0.667
Pelvic tilt (deg)	9.0 <2.0–15.0>	6.0 <−2.0–12.5>	−1.756	0.079
Pelvic range in sagittal plane (deg)	3.0 <2.0–5.0>	3.0 <2.0–5.0>	0.653	0.514
Pelvic obliquity (deg)	0.0 <−3.0–2.0>	0.0 <−3.0–3.5>	0.637	0.524
Pelvic range in frontal plane (deg)	6.5 <5.0–10.0>	6.0 <5.0–9.0>	−0.499	0.618
**Pelvic rotation (deg)**	**0.0 <−2.0–4.0>**	**2.0 <−2.0–5.0>**	**2.128**	**0.033**
Pelvic range in transverse plane (deg)	9.0 <6.0–14.0>	9.0 <5.0–14.5>	0.523	0.601
Hip flexion at initial contact (deg)	26.0 <16.0–35.0>	21.0 <15.0–33.5>	−1.269	0.204
Hip flexion at terminal stance (deg)	−13.0 <−20.0–−5.0>	−14.5 <−27.0–−3.5>	−0.556	0.578
Hip flexion in swing (deg)	28.0 <17.0–37.0>	25.5 <16.5–35.5>	−1.313	0.189
Pass retract (deg)	2.0 <0.0–5.0>	0.0 <0.0–5.0>	−0.241	0.810
Hip range in sagittal plane (deg)	37.5 <32.0–46.0>	38.0 <30.5–45.5>	0.227	0.820
Hip range in sagittal plane (%)	88.0 <76.0–105.0>	90.0 <77.5–108.5>	0.706	0.480
Hip abduction (deg)	0.0 <−5.0–5.0>	0.0 <−5.0–5.0>	−0.726	0.468
Hip range in frontal plane (deg)	10.0 <6.0–14.0>	10.0 <8.0–13.0>	0.819	0.413
Hip rotation (deg)	−8.5 <−30.0–12.0>	−5.0 <−15.0–15.0>	−0.608	0.543
Hip range in transverse plane (deg)	20.0 <14.0–30.0>	21.5 <12.5–43.0>	0.722	0.470
Knee flexion at initial contact (deg)	0.0 <−3.0–5.0>	1.0 <−5.0–5.0>	0.260	0.795
Knee flexion at weight acceptance (deg)	11.5 <5.0–20.0>	11.0 <5.0–16.5>	−0.268	0.789
Knee flexion at midstance (deg)	0.0 <−4.0–7.0>	2.0 <−4.5–9.0>	1.095	0.274
Max knee flexion at swing (deg)	53.0 <46.0–59.0>	53.0 <47.5–59.0>	0.053	0.958
Knee flexion in terminal swing (deg)	−5.0 <−10.0–−2.0>	−5.0 <−13.0–−3.0>	−1.323	0.186
Knee range in sagittal plane (deg)	55.5 <45.0–62.0>	54.0 <46.5–67.5>	0.016	0.987
Ankle flexion at initial contact (deg)	−5.0 <−10.0–0.0>	−3.0 <−12.0–0.0>	0.811	0.417
Max dorsiflexion in swing (deg)	14.0 <9.0–17.0>	13.0 <7.0–18.0>	−0.118	0.906
Max plantarflexion (deg)	−13.0 <−22.0–−2.0>	−17.0 <−28.5–−5.0>	−1.837	0.067
**Ankle range in sagittal plane (deg)**	**26.0 <21.0–35.0>**	**32.5 <24.0–39.5>**	**3.265**	**0.001**
Foot progression (deg)	−3.0 <−17.0–8.0>	−1.0 <−8.0–11.5>	1.099	0.272
**Range of foot progression (deg)**	**12.0 <8.0–19.0>**	**15.0 <10.0–25.5>**	**3.265**	**0.001**
FZ1 **	103.0 <93.0–117.0>	104.5 <95.0–122.0>	0.754	0.451
**FZ2 ****	**85.0 <71.0–94.0>**	**81.5 <68.0–90.0>**	**−2.101**	**0.036**
**FZ3 ****	**101.5 <94.0–101.0>**	**105.0 <97.5–116.0>**	**2.295**	**0.022**
FX1 **	10.0 <7.0–13.0>	9.0 <6.5–10.5>	−1.387	0.165
FX2 **	0.0 <0.0–2.5>	1.0 <0.0–3.8>	1.095	0.274
FY1 **	17.5 <12.0–22.0>	17.5 <11.5–24.0>	0.466	0.641
FY2 **	17.0 <12.0–23.0>	20.5 <12.0–25.0>	1.926	0.054

* normalized to age matched reference data of healthy children; ** normalized to body weight.

## Data Availability

The data presented in this study are available in anonymized form on request from the corresponding author. The data are not publicly available due to ethical reasons.

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
