# Peer review of "Effect of Plano-Valgus Foot on Lower-Extremity Kinematics and Spatiotemporal Gait Parameters in Children of Age 5–9"

_diagnostics, 2021, doi:10.3390/diagnostics12010002_

Round 1

Reviewer 1 Report

The manuscript presents a research study on plano-valgus foot in a group of children with clinically established FFF. The authors analysed, with different systems, the study population in two different conditions, static and dynamic. The results of statistical analyses show that a significant differences between healthy and flat feet was only found in the dynamic condition.

The manuscript is well structured and presents interesting results. Below I propose some suggestions to improve the quality and clarity of the work.

  1. It is often repeated in the text that the aim is to assess how flat feet affect spatiotemporal gait parameters. However, it seems that the aim is instead to evaluate a method of classifying this posture deformity. Please clarify this issue.
  2. It would be helpful if the analysis systems used were described in a little more detail. It would also be interesting if the authors could add some photos of the trials.
  3. Can the authors specify, in more detail, according to which criterion two different statistical tests were used? And I would also suggest specifying which test the results in Tables 2 and 3 refer to.
  4. I also suggest some small corrections to be made in the text: standardise Clark with Clarke; in the conclusion please correct the acronyms FF with the acronym FFF; the acronym ROM is missing.

Author Response

We would like to thank for all the comments and remarks. We hope that our paper is now improved. Below are point-to-point replies to all the reviewer’s remarks. All changes introduced during the review process are described, and marked with bolded font in the paper’s body.

  1. It is often repeated in the text that the aim is to assess how flat feet affect spatiotemporal gait parameters. However, it seems that the aim is instead to evaluate a method of classifying this posture deformity. Please clarify this issue.

From the cited literature and our own data we believe that the flat foot can negatively affect the gait pattern, both spatio-temporal and kinematic parameters. Taking into consideration the number of methods used to establish the diagnosis of the flat foot (there are even no agreed clinical criteria to do so) and criteria used it is difficult to describe what the flat foot really is. Therefore we do not think that we evaluate the methods of classification (there is no gold standard to which we can compare our patients) but we show, that the level of abnormality of the gait pattern can depend on the classification method within the same group of patients with clinical problem. Both methods: inkprinting with Clarke’s classification while standing, and Arch Index calculated from gait data from pedobarography are recognized methods, thus they give us different results. 

  1. It would be helpful if the analysis systems used were described in a little more detail. It would also be interesting if the authors could add some photos of the trials.

Fig.2 shows emed Novel system and screen of the VICON system with cameras in the background. We did not collected the consent of the patients’ guardians to use the photos showing them, therefore only such simplified presentation of the system is possible.

  1. Can the authors specify, in more detail, according to which criterion two different statistical tests were used? And I would also suggest specifying which test the results in Tables 2 and 3 refer to.

The following sentence was added at the beginning of the Results section: “The comparisons of the parameters between left and right leg, done with Wilcoxon signed-rank test showed no differences, thus the data from left and right legs were pooled together.” The captions of the Tables 2 and 3 are supplemented by: “Z – the results of the Mann-Whitney U test, p-level – probability value.”

  1. I also suggest some small corrections to be made in the text: standardise Clark with Clarke; in the conclusion please correct the acronyms FF with the acronym FFF; the acronym ROM is missing.

The suggested changes were made. We hope we did not miss any place in the text.

Reviewer 2 Report

The manuscript is focused on a common posture deformity among children, the plano-valgus foot which causes impairments in static posture and walking patterns. The authors use specific objective methodology to compare two classifications of flexible flat foot (FFF), finding that the static posture analysis does not help a clinician to find patients who have real functional walking problems. The topic is relevant to the journal's themes, the paper is well thought with a well-organized structure. Therefore it is worthy of publication, I just propose the following minor comments, that can be addressed by authors to improve the quality of the paper.

  • Authors present a thorough examination of the state of art relating to kinematic analysis of patients with flexible flat foot. All the reported literature deals with clinical assessment made in dynamic walking conditions. Is there any specific study performed on FFF children in static conditions? Is the static analysis of footprints a common method used on FFF patients in clinical or scientifical environment?
  • The chi-squared statistical test was used to check the distribution of data, but its results were not commented on. The Wilcoxon test and the Mann Whitney U test are two different non-parametric tests: the former is for non-independent samples, the latter for independent samples. According to which criteria were they used in the analyses?
  • Where did the authors select the reference values of spatio-temporal gait parameters to normalize acquired data?

Author Response

We would like to thank for all the comments and remarks. We hope that our paper is now improved. Below are point-to-point replies to all the reviewer’s remarks. All changes introduced during the review process are described, and marked with bolded font in the paper’s body.

  • Authors present a thorough examination of the state of art relating to kinematic analysis of patients with flexible flat foot. All the reported literature deals with clinical assessment made in dynamic walking conditions. Is there any specific study performed on FFF children in static conditions? Is the static analysis of footprints a common method used on FFF patients in clinical or scientifical environment?

In the cited literature the flat foot diagnosis is based either on clinical examination (in some papers the criteria of such examination were not given) during standing or on some kind of pedobarographic platform, but during standing, i.e. static condition. The most commonly used method to identify flat foot is clinical examination. In our study all children were clinically diagnosed with flat foot by physical and physiotherapist based on clinical criteria and measurements (this examination is described in Clinical examination subsection – we added the word “independently” to strengthen the fact that both specialists had the same opinion). In some studies the flat foot diagnosis was based on shape of the foot imprint, but during static condition. For example in all three studies by Lopez and co-workers added during the review process the Arch Index, on which the diagnosis of flat foot was based, was calculated from imprints during standing.

  • The chi-squared statistical test was used to check the distribution of data, but its results were not commented on. The Wilcoxon test and the Mann Whitney U test are two different non-parametric tests: the former is for non-independent samples, the latter for independent samples. According to which criteria were they used in the analyses?

The following sentence was added at the beginning of the Results section: “The comparisons of the parameters between left and right leg, done with Wilcoxon signed-rank test showed no differences, thus the data from left and right legs were pooled together.” The captions of the Tables 2 and 3 are supplemented by: “Z – the results of the Mann-Whitney U test, p-level – probability value.” The following sentence was added at the end of the Results section (just before Table 2): “As all the parameters were non-normally distributed (as showed by the results of the chi-square test) the data were summarized by the medians and 10th and 90th percentiles.”.

  • Where did the authors select the reference values of spatio-temporal gait parameters to normalize acquired data?

Appropriate reference was added.

Reviewer 3 Report

The topic is one of importance given the high number of presentations to health services that are related to concerns on  
the prevalence and related factors of plano-valgus foot in the  children population. Also, this is an interesting aim with the investigate assess see how a definition of the flexible flat foot (FFF) does influence the results of gait evaluation in a group of 49 children with clinically established FFF . I think it would be a more clear study if the following parts were revised and supplemented. These will be discussed below relative to the information of the manuscript.

General Comments:
Overall the manuscript is generally well written and is a topic of interest. However after reading it a number of times I am still left without key take-home points. I believe these points are in the results they just need to be developed.

Specific comments:
Abstract:
1) The authors state they will  assess  how a definition of the flexible flat foot (FFF) does influence the results of gait evaluation in a group of 49 children with clinically established FFF. This seems like too much of an over simplification of what was done. I do feel that it would be beneficial to explain what specifically you are looking at in relation to  plano-valgus foot(this also applies to the main body of the paper). Is it the development of lateral ankle sprain literature. This needs to be made clearer throughout the paper. (Major Compulsory Revision)

Introduction
2) The first paragraph should have a sentence or two added that better outlines why this study is important related with  plano-valgus foot and  impact of foot arch height on quality of life
https://pubmed.ncbi.nlm.nih.gov/25767305/ and https://pubmed.ncbi.nlm.nih.gov/30041462/

Furthemore, the authors do a poor job on reviewing relevant literatura related with importance with foot problems . Please revise the research of Navarro Flores  et al related with this question https://pubmed.ncbi.nlm.nih.gov/34267276/

3) In the last paragraph, the significance of the proposed word should be included highlighting why your work is important. what is the scientific contribution of this paper? it is not clear how this paper can make a significant contribution to the state of the art. (Major Compulsory Revision).
In addition, author´s hypotheses should be included and to change teh flow chart of the figure one for the method section.

5) This methods section is poor, needs to present a better rationale for the study and the methodology employed. Also, neither appear information related with inclusion and exclusion criteria, dates, protocol. The study design is a experimental research of ramdom sampling method, where the study was conducted in the hospital or in the university center? This research adhere to reporting STROBE guidelines? (Major Compulsory Revision).

6) Where the experiments carried out? In a hospital? In an educational institution? Provide this information.

7) Add figure 1 as a study flow chart for the readers. (Major Compulsory Revision).
8) Include p-values in all the tables (Major Compulsory Revision).
9) The Discussion section is a rehashing of the results. It does not appear that the authors include much interpretation of what the study findings mean for clinical practice or research. (Major Compulsory Revision)

FInally, the conclusión is weak and too long. (Major Compulsory Revision)

Author Response

We would like to thank for all the comments and remarks. We hope that our paper is now improved. Below are point-to-point replies to all the reviewer’s remarks. All changes introduced during the review process are described, and marked with bolded font in the paper’s body.

1) The authors state they will  assess  how a definition of the flexible flat foot (FFF) does influence the results of gait evaluation in a group of 49 children with clinically established FFF. This seems like too much of an over simplification of what was done. I do feel that it would be beneficial to explain what specifically you are looking at in relation to  plano-valgus foot(this also applies to the main body of the paper). Is it the development of lateral ankle sprain literature. This needs to be made clearer throughout the paper.

We hope that the changes introduced during the review process made our paper more clear.

Introduction
2) The first paragraph should have a sentence or two added that better outlines why this study is important related with  plano-valgus foot and  impact of foot arch height on quality of life https://pubmed.ncbi.nlm.nih.gov/25767305/ and https://pubmed.ncbi.nlm.nih.gov/30041462/ Furthemore, the authors do a poor job on reviewing relevant literatura related with importance with foot problems . Please revise the research of Navarro Flores  et al related with this question https://pubmed.ncbi.nlm.nih.gov/34267276/

The following sentence was added to the end of the Introduction section” “Additionally a study of Lopez and co-workers [22] found that the foot arch height has a global, negative impact on the quality of life of the schoolchildren, proving the importance of the foot deformities on the overall wellbeing. The similar study done under the same leadership [23] in the adults did not show any dependence between the height of the foot arch and quality of life, although another study performed in the adults with foot pathologies showed, that they have a worse quality of life than the general population [24].”.  All three suggested papers were used, and added to the literature.

3) In the last paragraph, the significance of the proposed word should be included highlighting why your work is important. what is the scientific contribution of this paper? it is not clear how this paper can make a significant contribution to the state of the art. In addition, author´s hypotheses should be included and to change teh flow chart of the figure one for the method section.

The following sentences were added at the end of the last paragraph of the Introduction: “The definitions of FFF used in practice differ from each other, which means that the applied method of foot classification influences the assessment of the patient's gait stereotype and the resulting therapeutic management. The importance of the research undertaken is due to the potentially negative impact of foot deformation on the quality of life in adulthood.”

5) This methods section is poor, needs to present a better rationale for the study and the methodology employed. Also, neither appear information related with inclusion and exclusion criteria, dates, protocol. The study design is a experimental research of ramdom sampling method, where the study was conducted in the hospital or in the university center? This research adhere to reporting STROBE guidelines? (Major Compulsory Revision). 6) Where the experiments carried out? In a hospital? In an educational institution? Provide this information.

The inclusion and exclusion criteria are given in the Materials and Methods section, subsection Patients, page 3. The information where the study was conducted was also given in the same paragraph. The Children’s Memorial Health Institute in Warszawa, Poland, is the biggest paediatric hospital in Poland, and research institute in paediatric medicine in one (https://nauka.czd.pl/en/, CZD or IPCZD are our acronyms in Polish).  We supplemented the study description in this section to clarify the doubts. All changes are marked with bolded text.

7) Add figure 1 as a study flow chart for the readers

The flow chart of the study was added as Fig.1 at the beginning of the Methods subsection.

8) Include p-values in all the tables

The data in Table 1 summarize the demographical data of the patients’ group. The comparisons are presented in Tables 2 and 3, where p-values were given, but the captions in the columns were p. They were changed to p-values. The captions of the Tables 2 and 3 are supplemented by: “Z – the results of the Mann-Whitney U test, p-level – probability value.”

9) The Discussion section is a rehashing of the results. It does not appear that the authors include much interpretation of what the study findings mean for clinical practice or research. FInally, the conclusión is weak and too long.

The following sentence was added at the end of the paper: “One of the main findings from our study is that the gait pattern pathology seen in the gait parameters can depend on the classification method within the same group of patients with clinical problem.”

Round 2

Reviewer 3 Report

I thank the corresponding author for their comments. I have read through the subsequent changes made to the manuscript and I have no further comments or suggestions.